# First-principles calculation of adsorption of cadmium and lead on the surface of a 1T-MnO$_2$ monolayer

**Julieth V. Dita-Casiano[1], Mario L. Arteaga-Calderón[1], Gladys R. Casiano-Jiménez[1], César Ortega-Lopez[1]\*, Miguel J. Espitia-Rico[2]\***

**1** Grupo Avanzado de Materiales y Sistemas Complejos GAMASCO, Laboratorio de Caracterización de Materiales, Universidad de Córdoba, Montería, Colombia, **2** Facultad de Ciencias Matemáticas y Naturales, Grupo GEFEM, Programa Académico de Física, Universidad Distrital Francisco José de Caldas, Bogotá Colombia

\* cortegal@correo.unicordoba.educ.co (COL); mespitiar@udistral.edu.co (MJER)

## Abstract

Manganese dioxide is a material with varied and interesting applications, especially including energy storage, the elimination of organic contaminants, and the removal of toxic atoms and molecules that pollute the environment. In this paper, the adsorption of toxic atoms of cadmium (Cd) and lead (Pb) on the surface of a 1T-MnO$_2$ monolayer is investigated using first-principles calculations. The calculated values of the adsorption energy show that the most energetically favorable site for the Cd and Pb atoms is P$_{Mn}$. This occurs when a Cd or Pb atom is located just above a Mn atom, with adsorption energies of −0.883 eV and −5.918 eV, respectively. The charge transfer in the 1T-MnO$_2$ adsorbate/monolayer interaction is determined via the Bader charge. Additionally, the thermodynamic, mechanical, and dynamic stability of a pristine 1T-MnO$_2$ monolayer is determined through calculations of the formation energy, elastic constants, and dispersion curves of the phonon density of states and the band structure, respectively. Our study suggests that the 1T-MnO$_2$ monolayer is a good material for adsorption of the toxic heavy metal atoms of Cd and Pb.

## 1. Introduction

In nature, manganese atoms exist in valence states $+2, +3, +4, +6$, or $+7$, so manganese has several oxidation states, which give rise to different manganese-based oxides such as MnO, MnO$_2$, MnO$_3$, Mn$_2$O$_3$, Mn$_2$O$_7$, etc., with MnO$_2$ being one of the most abundant. Manganese dioxide (MnO$_2$) in volume is a layered material formed by the stacking of monolayers of MnO$_2$ joined together by Van der Waals forces [1]. This material, which has been studied extensively due to its excellent physical and chemical properties, has technological applications in supercapacitors [2], sensors [3], catalysis [4], energy storage [5], and electrodes in lithium batteries [6]. These

**Data availability statement:** All relevant data are within the paper.

**Funding:** The GAMASCO Research Group wishes to express its thanks to the Universidad de Córdoba for the allocation of the physical space for Laboratorio de Cartacterización de Materiales where all the calculations were performed and which is located in the third level of the Facultad de Ciencas Básicas.

**Competing interests:** The authors have declared that no competing interests exist.

applications include $MnO_2$ by volume [7], thin films [8], nanosheets [9,10], surface areas [11], monolayers [12,13], and heterostructures [14–17], among others.

The $MnO_2$ monolayer crystallizes in the trigonal (denoted 1H) phase with $D_{3h}$ point group (in the Hermann-Mauguin notation $D_{3h} = \overline{6}m2$, where $\overline{6}$ : is called the rotation-inversion axis (improper rotation) and consists of a rotation of order 6 ($2\pi/6$ rad = $\pi/3$ rad = 60°) followed by a $\overline{1}$ inversion equivalent to a plane of symmetry or mirror, m is the vertical plane of symmetry (parallel to the main axis), and 2 is the rotation axis of order 2 ($2\pi/2$ rad = $\pi$ rad = 180°) perpendicular to the principal axis and the octahedral (denoted 1T) with $D_{3d}$ point group (in the Hermann-Mauguin notation $D_{3d} = \overline{3}m$), where $\overline{3}$ is called the rotation-inversion axis (improper rotation) and consists of a rotation of order 3 ($2\pi/3$ rad = 120°) followed by an inversion $\overline{1}$ (equivalent to a plane of symmetry or mirror), and m the symmetry plane (diagonal)). The $1T-MnO_2$ phase is the most stable [1,18]. Experimental studies show that $MnO_2$ in bulk is a good material for the adsorption of organic pollutants in wastewater [19], while the monolayer $1T-MnO_2$ adsorbs 99.4% of the tetracycline contained in wastewater produced by the pharmaceutical industry [20]. Recent experimental studies reveal that the manganese dioxide monolayer is an excellent material for the removal of atoms, molecules, and gases that are very dangerous global pollutants due to their high toxicity. Wu *et al.* [21] demonstrated that sulfur dioxide ($SO_2$) is efficiently captured via heterogeneous oxidation into sulfate on the surface of manganese dioxide ($MnO_2$) with a capture efficiency of nearly 100%. Liu *et al.* [22] developed a filter to capture $SO_2$ produced in the combustion of fossil fuels released into the environment by vehicles and industrial smokestacks. Peng *et al.* [23] demonstrated the great potential of a $MnO_2$ monolayer for the electrocatalytic transformation of $CO_2$ to valuable chemical products, because they proved that manganese dioxide reduces $CO_2$ to CO with an efficiency of 71%. This is of great importance for the chemical industry, because it is an alternative path for the synthesis of important chemical feedstocks and complex carbon-based fuels. On the other hand, theoretical studies predict the use of the $MnO_2$ monolayer in different areas. Deng *et al.* [24], using first-principles calculations, show that it exhibits good performance for Li storage capacity and diffusion, and therefore the monolayer is a promising electrode material for high-capacity Li ion batteries. Chen *et al.* [25], using density functional calculations, studied the adsorption of 5-fluorouracil (5-FU) on the surface of MnO2 and predicted that the monolayer is a good material for targeted delivery of drug molecules hiking on nanomaterials. Additionally, other theoretical studies based on the framework of density functional theory have focused on the adsorption of toxic atoms and molecules that cause environmental pollution. Independently, Zhu *et al.* [26] and Li *et al.* [27] studied the adsorption of NO and $O_2$ molecules on the $MnO_2$ surface, finding that this material is an excellent candidate for reducing emissions of these gases. Zhen *at al.* [28] calculated the adsorption of elemental mercury ($Hg^0$) on $MnO_2$, while Zhang *et al.* [29] studied the adsorption mechanism of mercury species (HgO, HgCl, and $HgCl_2$) on the $MnO_2$ surface. These theoretical investigations predicted that $MnO_2$ is a good material for the adsorption and oxidation of $Hg^0$ and mercury species HgO, HgCl, and $HgCl_2$, and therefore $MnO_2$ is a very promising material for the construction of devices (filters) that allow

reducing emissions of mercury and mercury species to the environment that result from the burning of fossil fuels in power plants and vehicle exhausts [30].

Environmental pollution is produced by various sources, one of which arises as a consequence of rapid industrial development, which leads to the release of toxic heavy metals such as Pb and Cd, which ultimately affect the environment. This release of toxic heavy metals has increased rapidly, causing much concern in the scientific community, since lead (Pb) and cadmium (Cd) are not easily degraded and have a high potential for bioaccumulation, which could cause damage to all living beings through the food chain [31,32]; therefore, the elimination or reduction of the emission of Pb and Cd into the environment is necessary and a priority. For this reason, in the present study we carried out a detailed investigation of the capture of toxic heavy metal ions such as Pb and Cd on the surface of the $MnO_2$ monolayer.

## 2. Computational details

First-principles total energy calculations were performed in order to study the adsorption of toxic atoms such as Pb and Cd onto the 1T-$MnO_2$ monolayer in the 3x3 geometry. Furthermore, the structural, energetic, and electronic properties of a pristine 3x3 1T-$MnO_2$ monolayer were studied. The calculations were executed within the framework of density functional theory (DFT) [33,34], implemented in the Quantum-ESPRESSO code [35]. The effects of correlation and exchange between electrons are included with the generalized gradient approximation (GGA), within the Perdew-Burke-Ernzerhof (PBE) parameterization [36].

For the calculations of the adsorption of an atom of Cd or Pb onto the surface of the 3x3-$MnO_2$ monolayer in its trigonal phase (space group P3m1, #164), a periodic slab was constructed. This slab contained a monolayer with three planes (two planes of oxygen and one of manganese) and an empty region of 26 Å, so that there would be no interactions between the periodic slab and its image in the z-direction. An 8x8x1 k-point mesh of was used, centered on Γ and generated according to the Monkhorst-Pack scheme [37]. The Van de Waals interactions between heavy metals (Cd and Pb) and the monolayer were considered using the semiempirical Grimme DFT-D2 correction [38]. The wave functions of the electrons expand into a set of plane waves [39], with a cut-off energy of 50 Ry. For the charge density, a cutoff energy of 500 Ry was used. The processes of geometric and structural optimization of systems with and without atomic adsorbates was achieved when the convergence criteria for the forces and energies of $10^{-3}$ Ry/Bohr and $10^{-4}$ Ry, respectively, were satisfied, and all calculations were performed with spin polarization.

The dynamic stability of the 1T-$MnO_2$ monolayer was established through the calculation of the phonon band structure and phonon density of states. This was carried out by means of a GGA (PBE) approach implemented in the Quatum-Espresso code and ultrasoft pseudopotentials using the Phonopy computational package [40]. The phonon band structure was calculated on the path M K with a total of 600 k-points per segment.

The mechanical stability was determined through the calculation of second-order elastic constants (SOEC), Young's modulus, and Poisson's ratio. The calculation of the SOEC of the 1T-$MnO_2$-(2x2) monolayer was performed using the ElaStic program [41] as the main tool, together with the Quatum-Espresso computational package with the GGA (PBE) approach. A maximum deformation of 2% (harmonic zone) was performed, with steps of 0.002.

## 3. Results and discussion

### 3.1. Pristine 1T-$MnO_2$ monolayer

#### 3.1.1. Structural properties of the 1T-$MnO_2$ monolayer.
To establish the structural properties of the 1T-$MnO_2$ monolayer in the 3x3 geometry with spatial group P-3m1 (#164), total energy calculations were carried out, which involved the geometric and structural optimization of the system. In this geometry, the monolayer has a total of 21 atoms: 9 of manganese and 18 of oxygen. For the geometric and structural optimization of the monolayer, the vc-relax calculation mode was used, where the energy and the thermodynamic stability of the monolayer is established by calculating the cohesion energy and the formation energy, respectively.

Fig 1 shows side and top views of the 1T-3x3 $MnO_2$ monolayer. It is made up of three planes of atoms, a top and a bottom plane of oxygen atoms and an intermediate plane of Mn atoms, as shown in Fig 1(a). In addition, the bond length $l_1 = l_2$ between the Mn and oxygen atoms (Fig 1 (b)) and the length $l_{12}$ are shown, which correspond to the vertical separation distance between the planes of upper and lower oxygen atoms that defines the thickness of the monolayer (Fig 1(a)).

Table 1 lists the values of the parameters obtained that characterize the structural and geometric relaxation, cohesion energy, and formation energy of the monolayer calculated in this investigation, along with values reported in literature.

## a)

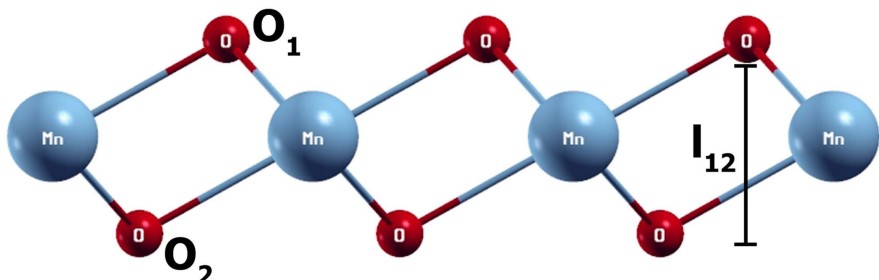

## b)

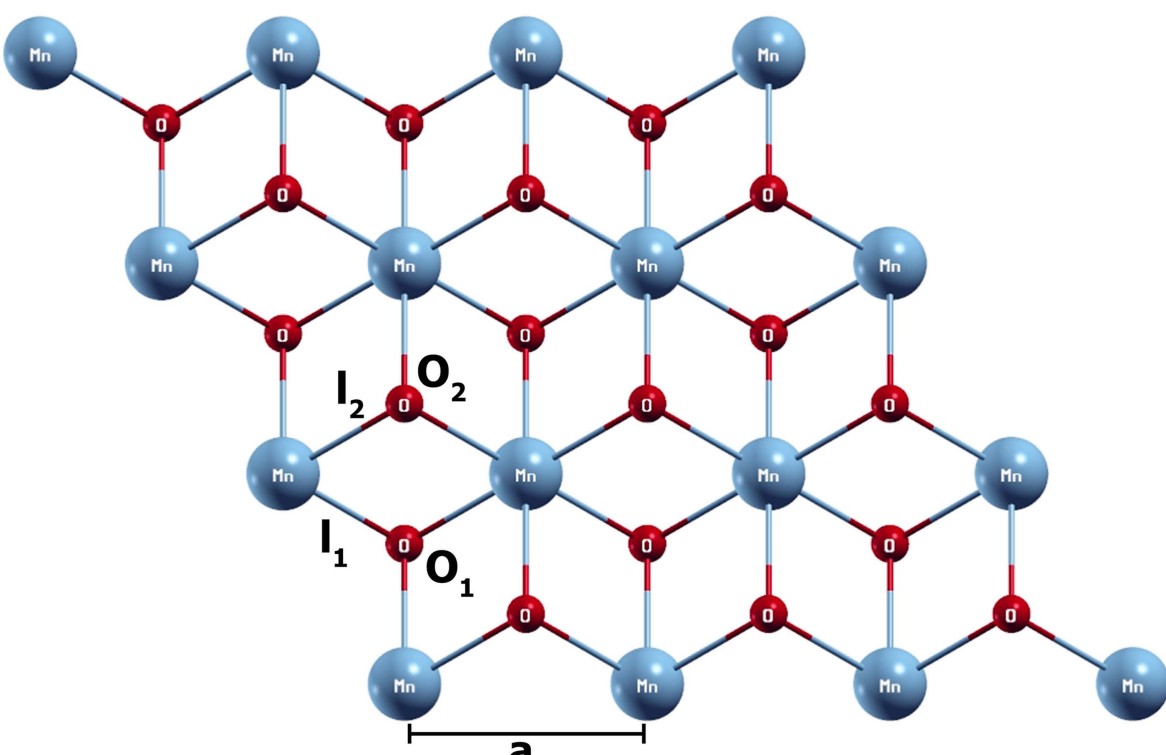

**Fig 1. a) Top view of the 1T-3x3 $MnO_2$ monolayer, b) Side view of the 1T-3x3 $MnO_2$ monolayer.**

**Table 1. Structural parameters and cohesion and formation energy of the T-MnO$_2$ monolayer, where a is the lattice constant, l is the bond distance between the manganese and oxygen atoms, l$_{12}$ is the thickness of the monolayer (oxygen-oxygen vertical distance), E$_c$ is the cohesion energy, and E$_f$ is the formation energy.**

| Sources | Method | a(Å) | l(Å) | l$_{12}$(Å) | E$_{coh}$(eV) | E$_f$(eV) |
|---|---|---|---|---|---|---|
| This study | GGA+U (PBE) | 2.946 | 1.950 | 1.907 | −23.775 | −5.741 |
| Morinson, J. *et al.* [44] | GGA-PBE | 2.852 | 1.920 | – | −15.670 | −5.310 |
| M. Kan, J. Zhou *et al.* [12] | GGA-PBE | 2.925 | 1.910 | – | – | – |
| Jana *et al.* [45] | Experimental | 2.920 | 2.030 | – | – | – |

In order to determine the thermodynamic and energetic stability of the monolayer, the formation and cohesion energies of the T- MnO$_2$ compound were calculated.

The formation energy is defined by [42]:

$$E_{Form} = E_{T-Bulk}^{MnO2} - \sum_i n_i \mu_i \qquad (3.1)$$

where $E_{T-Vol}^{MnO2}$, is the total energy of MnO$_2$ by volume in the structural phase considered, n$_i$ represents the number of atoms of species i, and μ$_i$ is the chemical potential of each individual atom i in its ground state, while the cohesion energy is defined by [43]:

$$E_{Coh} = E_{T-Bulk}^{MnO2} - \sum_i n_i E_i \qquad (3.2)$$

where $E_{T-Vol}^{TiO2}$ is the total energy of MnO$_2$ in volume in the structural phase considered, n$_i$ represents the number of atoms of species i, and E$_i$ is the total energy of the isolated individual atoms i.

We found that the values for the lattice constant (2.946 Å), the Mn-O bond length (1.950 Å), and the separation between the oxygen planes (1.907 Å) are in excellent agreement with those reported in the literature. The values for the formation energy and cohesion energy were −23.775 eV/cell and −5.71 eV/cell, respectively. The value of the cohesion energy reported in the present study is higher in absolute value than that reported by Morrison [44], while the value of the formation energy is quite close to that reported by the same authors. Additionally, we found that the formation energy is negative; therefore, the monolayer is thermodynamically stable (in this process, an exothermic reaction occurs), that is, its determination is experimentally viable.

In this spatial group, we found the value of the exfoliation energy ($E_{exf}$) and binding energy ($E_b$), which are defined as [46]

$$E_{Exf} = \frac{E_{tot}(monolayer) - E_{tot/layer}(bulk)}{A_0} \qquad (3.3)$$

where $E_{tot}(monolayer)$ is the total energy of a monolayer, $E_{tot/layer}(bulk)$ is the total energy of a bulk per layer, and $A_0$ is the plane area of the equilibrium bulk, and

$$E_b = \frac{E_1 - \frac{1}{n}E_n}{A} \qquad (3.4)$$

where, $E_1$ and $E_n$ are the energy of the monolayer and of the bulk, respectively, $A_0$ is the area of the monolayer, and $n$ is the number of monolayers.

The value of the calculated exfoliation energy was 16.925 meV/Å, which is lower than that reported for graphene (21.00 meV/Å) [46]. From this result, it can be inferred that the 1T-MnO$_2$ monolayer can be extracted from the surface of the material in volume, which is in accordance with the criterion established by Björkman *et al.* [48], according to which mono-layers with exfoliation energies in the range of 15–20 meV/Å are easily exfoliated. Additionally, the value 16.925 meV/Å for 1T-MnO2 is close to reported values of the exfoliation energy for monolayers of other materials, such as VO$_2$ (16.250 meV/Å), CrO$_2$ (18.690 meV/Å), MoO$_2$ (19.210 meV/Å) [47], MgBr$_2$ (17.262 meV/Å), MgI$_2$ (15.808 meV/Å), MoS$_2$ (18.211 meV/Å), and VS$_2$ (17.071 meV/Å) [48], monolayers that are classified as easily exfoliable. Finally, the calculated interlayer binding energy is very close to the exfoliation energy, which is in excellent agreement with values reported for the energies of E$_b$ and E$_{exf}$ for other transition-metal dioxides [49].

**3.1.2 Electronic properties of the 1T-MnO$_2$ monolayer.** The electronic characterization of the 1T-MnO$_2$ monolayer was carried out by calculations of the band structure, the electronic density of state (DOS), and the Bader charge distribution.

Fig 2 shows the band structure of the 1T-MnO$_2$ monolayer.

From Fig 2, it can be seen that the monolayer exhibits a semiconductor behavior, with an indirect band gap of 2.08 eV for the spin-down component and 1.97 eV for the spin-up component. The monolayer has magnetic properties, which is reflected in the asymmetry of the spin-up and spin-down contributions, with a total magnetic moment of 3μ$_\beta$/cell.

Fig 3 shows the total and partial density of state (DOS) of the 1T-MnO$_2$ monolayer.

Fig 3 confirms the semiconductor character of the 1T-MnO$_2$ monolayer. The orbitals that make up most of the valence band near the Fermi level are the states Mn-4d and O-2p. Furthermore, it can be seen that the magnetic properties come from the hybridization of the p-d orbitals of the Mn and O atoms. Table 2 shows the results of the calculations of the Bader charge distribution of the 1T-MnO$_2$ monolayer.

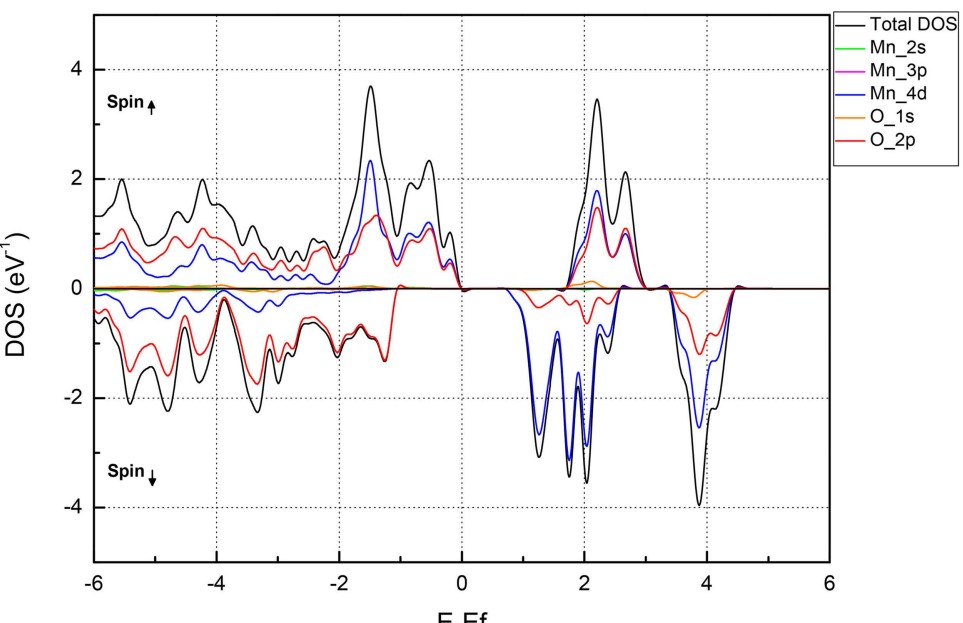

**Fig 2. Structure of spin-up (black line) and spin-down (red line) electronic bands for the 1T-MnO$_2$ monolayer.**

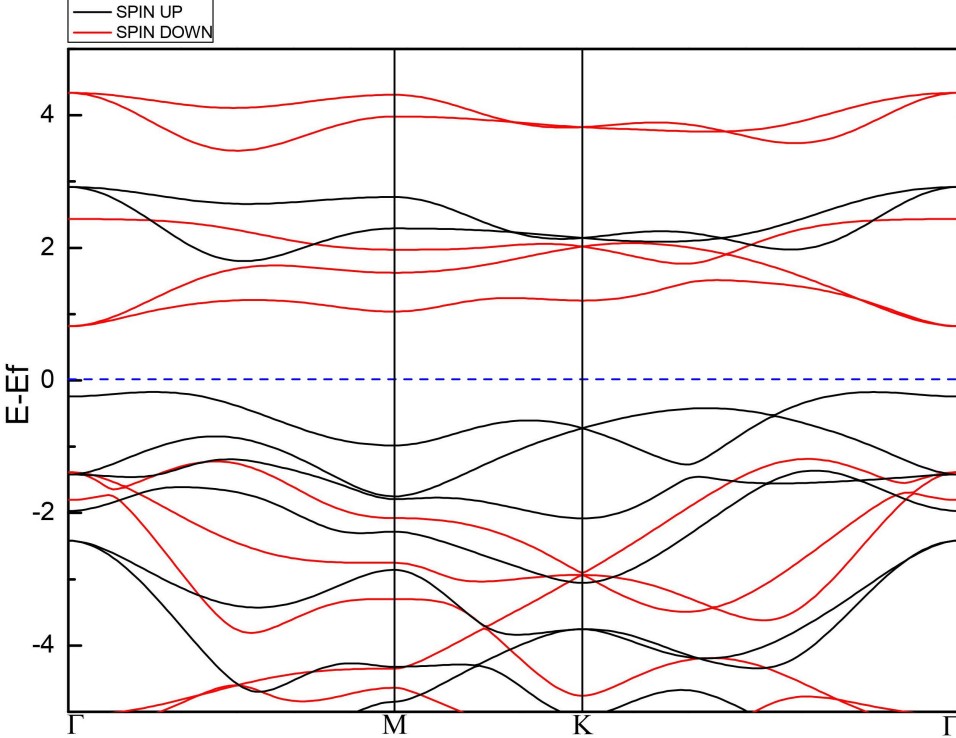

**Fig 3. Total and partial electronic state density (DOS) for the MnO$_2$ monolayer.**

**Table 2. Bader charge distribution of the 1T-MnO$_2$ monolayer, where Q is the charge of the free atom, Q\* is the charge of the bonded atom, ΔQ the charge variation, and n the atomic charge density.**

| Atom | Q | Q* | $\Delta Q$ | n |
|---|---|---|---|---|
| Mn | 15 | 13.141 | −1.859 | 0.245 |
| O1 | 6 | 6.928 | 0.928 | 0.007 |
| O2 | 6 | 6.929 | 0.929 | 0.007 |

From the results shown in Table 2, it can be inferred that there is an ionic bond in the 1T-MnO$_2$ unit cell, because the manganese atom loses approximately 2 electrons, which converts it into a positive ion, while each of the oxygen atoms gains approximately one electron, which converts them into negative ions.

From the results obtained, shown in Table 3, it can be seen that the distribution of charge by orbital is asymmetrical, as observed in the electronic DOS (Fig 3). The charge in the manganese atom is mainly distributed in the p and d orbitals, and the p orbital shows symmetry in its charge for the spin-up and spin-down components, while the d orbital has a higher charge for the spin-up component. On the other hand, in oxygen atoms, the charge is mainly distributed in the p-orbitals for the spin-up and spin-down components. In particular, these are the orbitals that mainly contribute to the electronic character of the monolayer. Fig 4 shows a side view of the contour map of the charge distribution in the 1T-MnO$_2$ monolayer.

Fig 4 shows the electronic localization function, visualized through the differential electron density Δn(r). In this representation, blue indicates regions of greater concentration of electronic charge, while reddish denotes areas of

**Table 3. Molecular charge distribution by orbital in the 1T-MnO$_2$ monolayer, with spin-up and -down components of spin polarization; Qs, Qp, and Qd represent the charge of the atom in each orbital, according to the spin polarization.**

| Spin | Up | | | Down | | |
|---|---|---|---|---|---|---|
| Orbital | Qs | Qp | Qd | Qs | Qp | Qd |
| Mn | 1.237 | 2.999 | 4.195 | 1.202 | 3.000 | 0.827 |
| O1 | 0.931 | 2.316 | 0.000 | 0.923 | 2.540 | 0.000 |
| O2 | 0.931 | 2.318 | 0.000 | 0.908 | 2.382 | 0.000 |

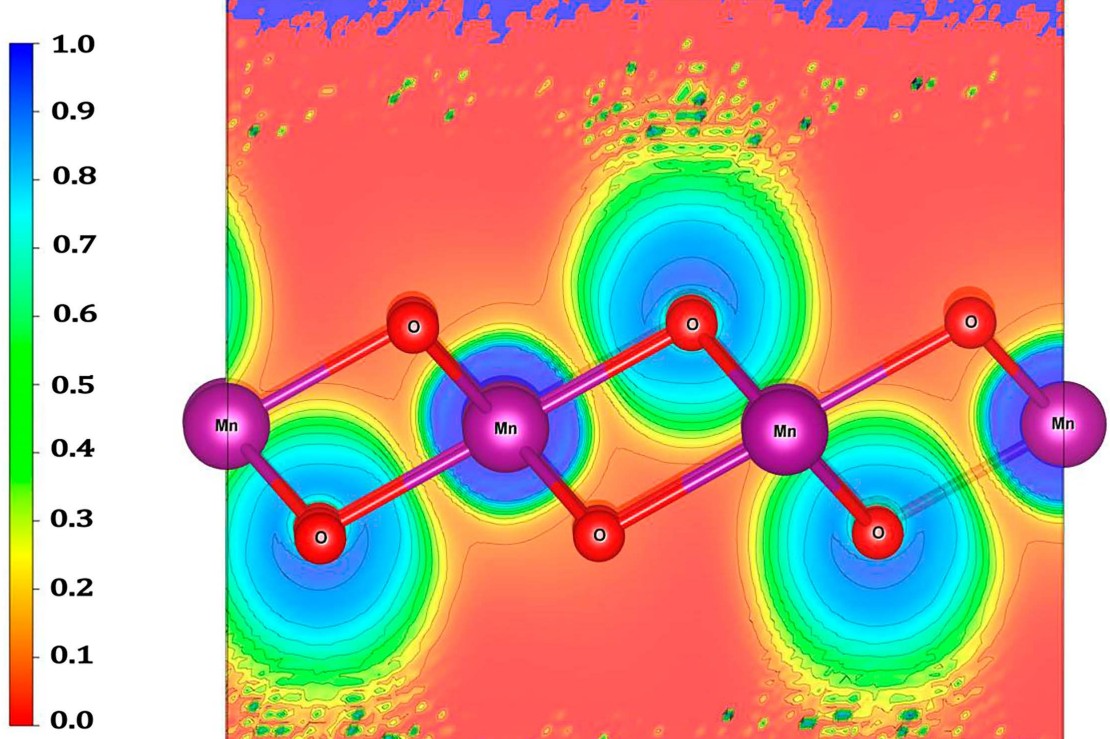

**Fig 4. Contour map of the charge distribution on the 1T-MnO$_2$ monolayer, side view.** The red area indicates little or no charge, and blue indicates high charge presence. The atoms of the top and bottom plane are of oxygen, and those of the central plane are of manganese.

less density or absence of net charge. The selected cutting plain specifically intersects with one of the manganese (Mn) atoms and oxygen (O) atoms above and below. The image in Fig 4 shows a classic pattern of an ionic bond: the oxygen exhibits high electronic density (azure) due to the charge transference from the manganese. On the contrary, the density observed around the manganese predominantly corresponds to its core and semi-core electrons, which remain virtually inaccessible for the oxygen. This result directly validates the calculations of the Bader charge, presented in Table 2.

### 3.1.3. Mechanical stability of the 1T-MnO$_2$ monolayer.

Study of the mechanical stability of the monolayer was carried out by calculating the elastic constants and applying the mechanical stability criteria for a hexagonal monolayer. Calculations were done for Young's modulus and Poisson's ratio, which are fundamental for the description of two-dimensional materials (Table 4).

**Table 4. Second-order elastic constants and linear mechanical properties of the 1T-MnO$_2$ monolayer in units of Nm$^{-1}$.**

|  | C$_{11}$ | C$_{12}$ | C$_{66}$ | Y | $\nu$ |
|---|---|---|---|---|---|
| This study | 132.2 | 25.0 | 53.6 | 127.4 | 0.19 |
| MoS$_2$ [50] | 132.7 | 33.0 | 49.85 | 124.5 | 0.25 |
| MoSe$_2$ [51] | 108.0 | 25.0 | – | 103.9 | 0.24 |
| WSe$_2$ [52] | 119.0 | 23.1 | – | 115.6 | 0.19 |

The calculations of the elastic constants were done with the methodology described in section 2.0, using the ElaStic program: performing a maximum strain of 0.02 (2%), with steps of 0.002. In the results of the calculations with the ElaStic program, the total energy of the system is expressed as a function of the deformations, which is a Taylor series expressed as:

$$E[\eta] = E_0 + \sum_{ij} \sigma_{ij}^{(0)} \eta_{ij} + \frac{1}{2} \sum_{ij,kl} C_{ij,kl}^{(2)} \eta_{ij}\eta_{kl} + \frac{1}{6} \sum_{ij,kl,mn} C_{ij,kl}^{(3)} \eta_{ij}\eta_{kl}\eta_{mn} + \cdots$$

In this case, in order to describe the elastic properties of two-dimensional systems, it is sufficient to calculate the second-order elastic constants (SOEC), which are described as:

$$C_{ij,kl}^{(2)} = \left. \frac{\partial^2 E[\eta]}{\partial_{ij}\partial_{kl}} \right|_{\eta=0}$$

The stability criteria for the 1T-MnO$_2$ monolayer in this phase are $C_{11} > 0$, $C_{66} > 0$, and $C_{11} > |C_{12}|$ and $C_{66} = (C_{11} - C_{12})/2$.

The values obtained for the monolayer are $C_{11} = 132.2\ nm^{-1}$, $C_{12} = 25\ nm^{-1}$, and $C_{66} = 53.6\ nm^{-1}$. It can be seen that $C_{11} > |C_{12}| > 0$ and $C_{66} > 0$.

Therefore, according to the mechanical stability criteria for two-dimensional systems, the 1T-MnO$_2$ monolayer is mechanically stable.

The calculation of Young's modulus was done via the equation:

$$Y = \frac{(C_{11}^2 - C_{12}^2)}{C_{11}}$$

For this system, we determined a Young's modulus of Y = 127.45 Nm$^{-1}$. The calculation of Poisson's coefficient was carried out with the equation

$$\nu = \frac{C_{12}}{C_{11}}$$

For this system, a Poisson coefficient of ν = 0.19 was determined.

The calculated values are similar to those reported for transition-metal dioxide systems are shown in Table 4 [45,46].

### 3.1.4. Dynamic stability of a 1T-MnO$_2$ monolayer.

The dynamic stability of the monolayer is determined by calculating the vibrational properties, which are established by the phonon band structure and the phonon density of states, as shown in Fig 5, left and right, respectively. The calculations were carried out with the Phonopy program as

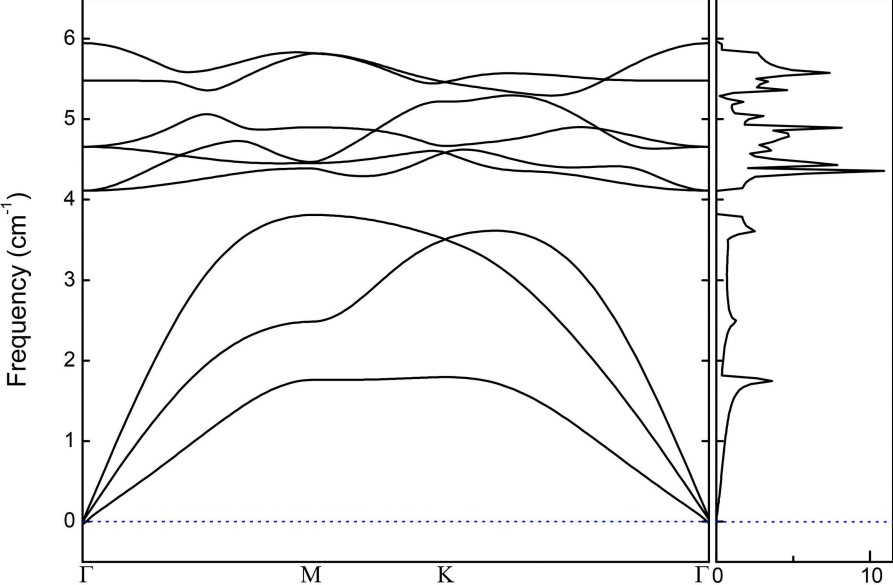

**Fig 5. Phonon band structure and phonon density of state of the 1T-MnO$_2$ monolayer.**

implemented in the Quantum Espresso code, using a 1T-MnO$_2$ 3x3x1 monolayer. The phonon band structure was calculated along the path Γ-M-K-Γ with a total of 600 k-points chosen.

From the results shown in Fig 5, it can be seen that there are no dispersion curves or phonon states below the zero of the frequencies; therefore, the monolayer 1T-MnO$_2$ is dynamically stable.

### 3.2. Cd and Pb adsorption

In this section, the adsorption of Cd and Pb onto the 1T-MnO$_2$ monolayer is examined. In particular, the most energetically favorable adsorption sites are determined, the structural properties are analyzed, and finally, the electronic properties of the adsorbate/monolayer system are studied through the Bader charge distribution and the charge differences isosurfaces.

**3.2.1. Structural properties of the adsorption of Cd and Pb onto the 1T-MnO$_2$ monolayer.** To determine the most favorable adsorption models of Cd and Pb onto the 1T-MnO$_2$-3x3 monolayer, we considered five (5) non-equivalent special sites: above an oxygen atom O (site $P_{O1}$): in the top plane oxygen atoms; above a manganese atom Mn (site $P_{Mn}$): in the intermediate plane of manganese atoms; above an oxygen atom O (site $P_{O2}$): in the bottom plane; above the $Br_1$ bridge (site $P_{B1}$): at the midpoint between an oxygen atom of the upper plane and an oxygen atom of the lower plane; and above $Br_2$ bridge (site $P_{B2}$): at the midpoint between an oxygen atom of the upper plane and a manganese atom of the intermediate plane, as shown in Fig 6.

The calculations of Pb and Cd adsorption were carried out following the methodology outlined in section 2.0, for which a periodic slab with a vacuum of 26 Å was used. The energy of adsorption was calculated by means of equation 3.5 [44,52]:

$$E_{ad} = E_{MnO_2}^{ad} - E_{MnO_2} - E_{ad}^{iso}$$

(3.5)

where $E_{MnO_2}^{ad}$ is the total energy of the 1T-MnO$_2$ monolayer plus the atomic adsorbate, $E_{MnO2}$ is the energy of the pristine monolayer, and $E_{ad}^{iso}$ is energy of the isolated adsorbate.

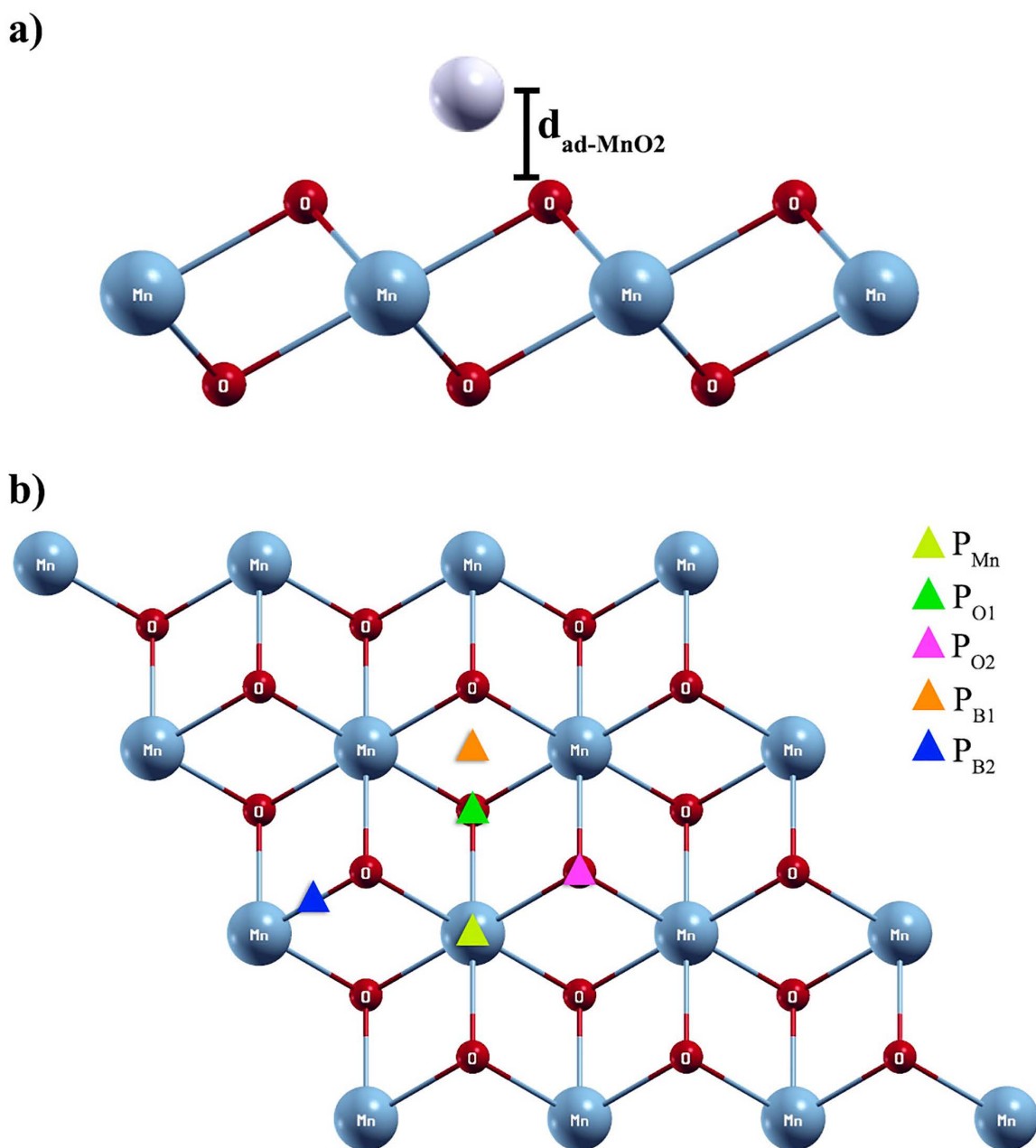

**Fig 6. a) Side view and b) special non-equivalent sites, on the surface of the 1T-MnO$_2$ monolayer for the adsorption of Pb and Cd.**

For each of the five adsorption sites investigated, Table 5 reports the corresponding adsorption energy and the perpendicular distance ($d_z$) from the Cd and Pb adatoms to the plain defined by the oxygen atoms of the top layer of the IT-MnO$_2$ monolayer. This distance is denominated the ad-monolayer bond length (($d_{ad-MnO2}$). It is crucial to point out that this parameter is distinct from the ad-O bond distance, which refers to the real bond length between the adatom and its closest oxygen atom.

**Table 5. Adsorption energies of Cd and Pb onto the surface of the 1T-MnO$_2$ monolayer at the non-equivalent sites considered in this study, $d_{Cd-MnO2}$ and $d_{Pb-MnO2}$, are the adsorbate-monolayer bond lengths.**

| Sites | Cd | | Pb | |
|---|---|---|---|---|
| | $E_{ads}(eV)$ | $d_{Cd-MnO2}$ (Å) | $E_{ads}(eV)$ | $d_{Pb-MnO2}$ (Å) |
| P$_{O1}$ | −0.756 | 1.492 | −5.085 | 2.156 |
| P$_{O2}$ | −0.869 | 1.671 | −3.693 | 1.704 |
| P$_{Mn}$ | −0.883 | 1.561 | −5.918 | 1.368 |
| P$_{B1}$ | −0.830 | 1.803 | −5.087 | 1.780 |
| P$_{B1}$ | −0.830 | 1.827 | −4.930 | 1.698 |

Table 5 shows that the most energetically favorable site for the Cd atom is PMn, with an adsorption energy of −0.883 eV and a Cd-MnO$_2$ bond length of 1.561 Å. The most energetically favorable site for the Pb atom is PMn, with an adsorption energy of −5.918 eV and a Pb-MnO$_2$ bond length of 1.368. In both cases, the most energetically favorable adsorption site is PMn; however, the bond strength in the Pb/MnO$_2$ monolayer system is higher than in the Cd/MnO$_2$ monolayer system. From these values, it can be inferred that in both cases, the adsorption is chemical.

While there is a general tendency for the chemical adsorption to indicate covalent (polar or non-polar) or ionic bonds and the physical adsorption to involve weaker van der Waals forces, there are exceptions. Analysis of the Bader charge together with the charge difference, which shows the transfer and distribution of the interaction charge between the atomic adsorbates, the surface, and the atoms pertaining to it, affords a more direct and reliable way to determine the type of bond.

For the Pb atomic ad-sorbate on the surface of the T-MnO$_2$ monolayer, the energetically most favorable site is PMn, which has an adsorption energy of −5.918 eV and a Pb-O bond length of 2.24 Å for each of the three(3) oxygen atoms located in the top plane of the T-MnO$_2$ monolayer (see Fig 7). The three (3) oxygen atoms located on top of the monolayer form an equilateral triangle centered on the PMn special site, where Mn is a manganese atom that belongs to the middle plane of the monolayer and is located just below the projection of the PMn special site. Furthermore, the bottom plane of the monolayer is formed by oxygen atoms, as is shown in Fig 7. From this figure, it can be seen that when Pb is adsorbed onto the PMn site, the Mn atom is displaced from its original position (middle plane of the monolayer) to a position slightly below the bottom plate of the monolayer. On finalizing the geometric optimization, the final separation distance of the Pb atomic adsorbate from the displaced Mn atom is 3.75 Å. Notwithstanding that a significant displacement of the Mn atom is seen, induced by the Pb atomic adsorbate, the geometric optimization is satisfactory, since the total pressure of the adsorbate-adsorbent system is 0.32 kba.

Likewise, for the Cd atomic adsorbate on the surface of the T-MnO$_2$ monolayer, the most energetically favorable site is also PMn, which has an adsorption energy of −0.883 eV and a Cd-O bond length of 3.32 and 3.22 Å for the oxygen atoms labelled (1, 2) and 3, located in the top plane of the T-MnO$_2$ monolayer, respectively (see Fig 8). From Fig 8, it can be seen that when Cd is adsorbed onto the PMn site, the Mn, which belongs to the middle plane of the monolayer and is located just below the PMn site, is displaced from its original position to the bottom plane of the monolayer. On finalizing the geometric optimization, the final separation distance of the Pb atomic adsorbate from the displaced Mn is 2.90 Å. Notwithstanding that a significant displacement of Mn is seen, induced by the Cd atomic adsorbate, the geometric optimization is satisfactory, since the total pressure of the adsorbate-adsorbent system is −0.02 kbar.

**3.2.2. Electronic properties of the adsorption of Cd and Pb onto the 1T-MnO$_2$ monolayer.** In this section, the electronic properties of the adsorption of the Cd and Pb atoms onto the surface of the 1T-MnO$_2$ monolayer are established by means of Bader charge calculations in order to determine how the charges are redistributed in the atoms of the monolayer after the adsorption of the atomic adsorbates is complete. This is extremely important, as demonstrated by the study by Ahlers *et al.* [53], who prove that charge transfer plays a crucial role in the intensity of the adsorption

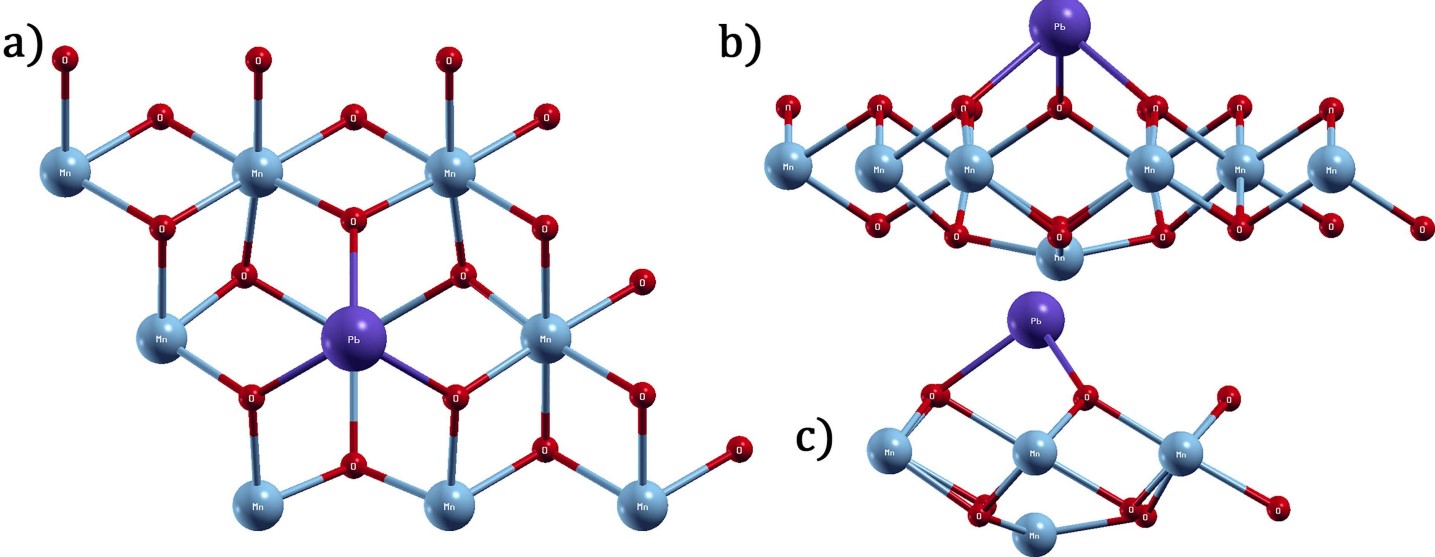

**Fig 7. MnO₂ monolayer after Pb adsorption, a) top view, b) side view, where slight local deformation can be seen, c) side view where it can be seen that the crystalline structure of the pristine monolayer is quasi-preserved.**

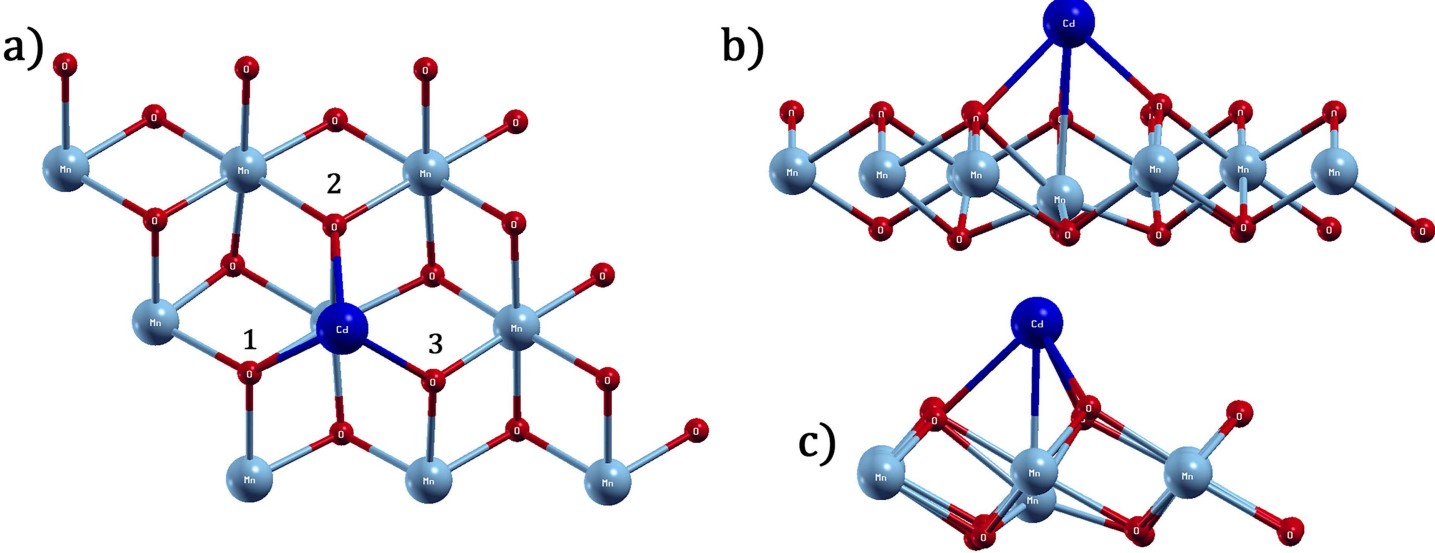

**Fig 8. MnO₂ monolayer after adsorption of Cd, a) top view, b) side view, where the slight local deformation is seen, and c) side view where it can be seen that the crystalline structure of the pristine monolayer is quasi-preserved.**

and resistance of the monolayer. In this way, what type of bond occurs in adsorbate/monolayer interactions can be established.

Fig 9 shows a top view of the T-MnO₂-3x3 monolayer. Here, the atoms of Mn from 1 to 9 and the atoms of O have been labelled from 1 to 18. Also, the most energetically favorable site (PMn) for the adsorption of an atom of Cd or Pb is shown.

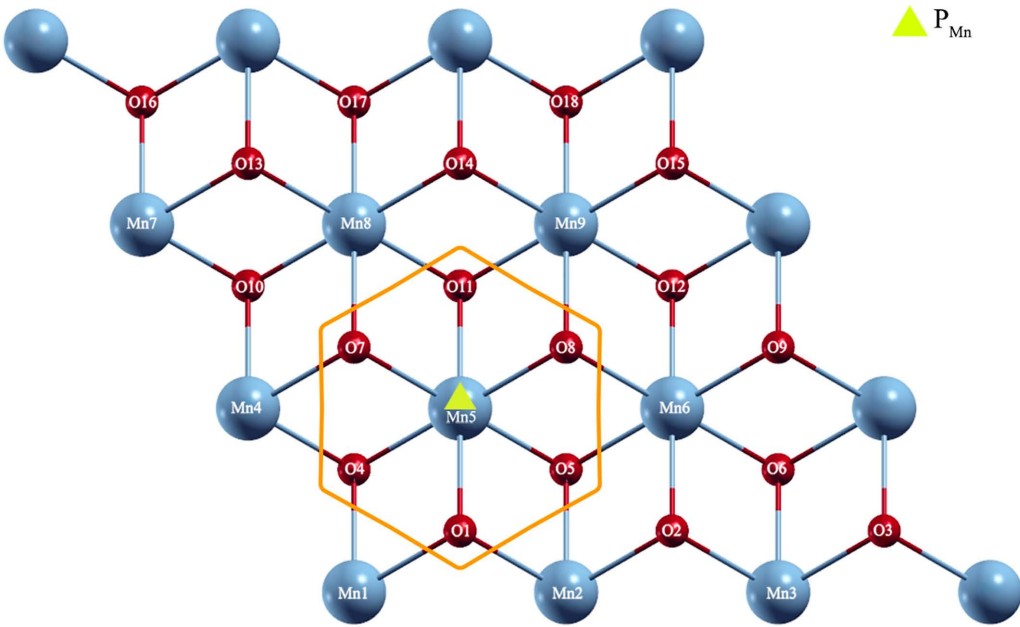

**Fig 9. Top view of the 1T-MnO$_2$-3x3 monolayer.** The orange hexagon shows the first and second neighbors of an atom of Cd or Pb adsorbed at the PMn site.

The PMn site and the first neighbors of the atom adsorbed onto PMn, i.e., the atoms $O_4$, $O_5$, $O_{11}$ (located in the upper plane) and $Mn_5$ (located in the intermediate plane), are enclosed in the orange hexagon, while the oxygen atoms $O_1$, $O_7$, and $O_8$ (located in the bottom plane) correspond to the second neighbors. The first and second neighbors of the corresponding atomic adsorbate are the main contributors to the calculations of the Bader charge, because they are the closest to the adsorbed atom.

Tables 6 and 7 show the charge distribution of the first and second neighbors of the PMn site before and after the adsorption of a Cd and Pb atom, respectively.

As seen in Table 6, once the adsorption process of a Cd atom onto PMn is completed, there is a charge redistribution in the first and second neighbors of the Cd adsorbate. It can be seen that for the second neighbors, $O_1$, $O_7$, and $O_8$ atoms, there is a slight increase in charge, while for the $O_4$, $O_5$ and $O_8$ atoms the increase in charge is~0.1e$^-$. The Mn atom undergoes the greatest increase in charge, with a value of ~ 0.2e$^-$.

**Table 6. Effects of Cd adsorption onto the surface of the 1T-MnO$_2$ monolayer on the charge of its first neighbors for the most energetically favorable site.**

| Atom | Q before | Q after | $\Delta Q$ |
|---|---|---|---|
| $Mn_5$ | 13.141 | 13.333 | 0.192 |
| $O_1$ | 6.927 | 6.959 | 0.032 |
| $O_4$ | 6.931 | 7.048 | 0.117 |
| $O_5$ | 6.931 | 7.054 | 0.124 |
| $O_7$ | 6.928 | 6.965 | 0.037 |
| $O_8$ | 6.929 | 6.960 | 0.031 |
| $O_{11}$ | 6.931 | 7.051 | 0.121 |

In Table 7, it can be seen that when a Pb atom is adsorbed at the P$_{Mn}$ site, a charge redistribution occurs in the first and second neighbors of the atomic adsorbate Pb, but unlike what happens for the adsorption of a Cd atom, in the adsorption of a Pb atom, in all oxygen atoms the same value of increase in charge of ~ 0.1e⁻ occurs, while for the Mn atom, the increase in charge is~0.4e⁻, which is twice the increase of that of the case of the adsorption of Cd. The charge differential transfer during the adsorption of Cd/Pb is quantified via a Bader analysis (~1,0 |e⁻| for Cd and ~1,4 |e⁻| for Pb)

Finally, to check the charge redistribution and the mechanisms of interaction between the monolayer and the adsorbate at the PMn site, the redistribution of the charge density Δρ, which is defined as follows, was calculated and plotted [53,54]:

$$\Delta\rho = \rho_{MnO_2}^{ad} - \rho_{MnO_2} - \rho_{ad}^{iso}$$

(3.6)

where $\rho_{MnO_2}^{ad}$, $\rho_{MnO_2}$, and $\rho_{ad}^{iso}$ represent the charge density of the 1T-MnO$_2$ monolayer with a Cd or Pb atom adsorbed at the PMn site, the charge density of the pristine monolayer, and the charge density of the isolated adsorbate, respectively. Figs 10 and 11 show the isosurfaces corresponding to the charge density variations for the 1T-MnO$_2$ monolayer with a Cd or Pb atom adsorbed at the most energetically favourable site. On the isosurfaces, yellow and blue indicate charge gain and loss, respectively.

**Table 7. Effects of Pb adsorption on the surface of the 1T-MnO$_2$ monolayer on the charge of its first neighbors for the most energetically favorable site.**

| Atom | Q before | Q after | $\Delta Q$ |
|---|---|---|---|
| Mn$_5$ | 13.141 | 13.531 | 0.390 |
| O$_1$ | 6.927 | 7.039 | 0.112 |
| O$_4$ | 6.931 | 7.063 | 0.132 |
| O$_5$ | 6.931 | 7.060 | 0.130 |
| O$_7$ | 6.928 | 7.038 | 0.110 |
| O$_8$ | 6.929 | 7.042 | 0.113 |
| O$_{11}$ | 6.931 | 7.062 | 0.131 |

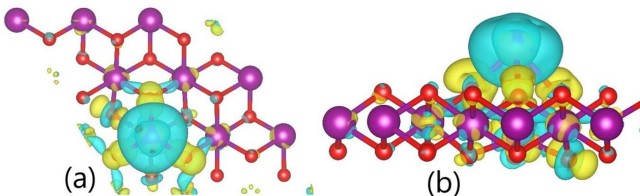

**Fig 10. Charge difference isosurfaces, in the adsorption of Cd on 1T-MnO$_2$ (8.a top view, 8.b view side).**

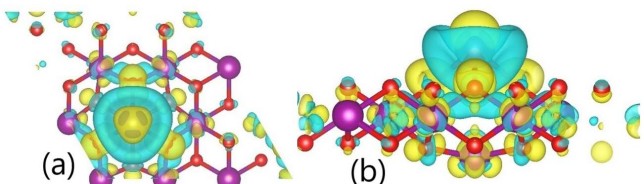

**Fig 11. Isosurfaces of the charge difference in the adsorption of Pb onto 1T-MnO$_2$ (9.a top view, 9.b view side).**

Figs 10 and 11 show the dynamics of the charge transfer when Cd or Pb are adsorbed onto the PMn site. In both cases, the charge is donated by the adsorbate principally to the first and second neighbors within the monolayer. The O4, O5, and O11oxygen atoms (first neighbors) and the Mn5 atom exhibit the most significant gain in charge, which corresponds to the regions of intense yellow visible in the figure.

In particular, the charge redistribution represented in Figs 10 and 11 underlines the hybrid nature of the bond. A blue surface (charge depletion) involves both adsorbates, which is reminiscent of an ionic bond. Nevertheless, within the substrate itself (the monolayer), the charge gain in the oxygen atoms is manifested as yellow rings in its peripheries, markedly contrasting with the blue zones in their centers. This visual pattern signifies a local charge redistribution within the monolayer. This redistribution, intrinsic to the structure of the substrate, is a distinctive characteristic of interactions with an emerging polar covalent component, going further than a purely ionic behavior.

This behavior is exemplified by the adsorption of Pb (Fig 11). Here, the charge donated by the Pb is distributed equally among the six neighboring oxygen atoms at the adsorption site (first neighbors O4, O5, and O11, and second neighbors O1, O7, and O8, shown in Fig 7) They coincide with the six distinct yellow spherical regions that surround the site in Fig 11. Simultaneously, the Mn5 atom, just below the adsorbate, gains the greatest quantity of charge, which corresponds to the small yellow zones around it (see Fig 11). The efficiency and the reach of this charge redistribution mechanism are key factors that explain the high sensitiveness of the IT-MnO$_2$ monolayer to the processes of adsorption of Cd and Pb.

Although the difference in the electronegativity (ΔEN) suggests a predominantly ionic character for Cd–O (ΔEN = 1.75) and a polar covalent character for Pb–O (ΔEN = 1.11), analysis of the Bader charge distribution reveals a more complex behavior. The transferred charges (~1.0 |e$^-$| for Cd and ~1.4 |e$^-$| for Pb) quantitatively satisfy the criterion of classical ionization. However, this charge donated by the metals is not localized in individual oxygen atoms, but rather is delocalized among the six oxygen atoms of the substrate (first and second neighbors). This redistribution adopts an extended lattice pattern that goes beyond the immediate neighbors and is stabilized by the structural disposition O$_{superior}$–Mn–O$_{inferior}$ (Fig 4). This delocalization contradicts the expected localized geometry for discrete ionic bonds. As a consequence, the sorbent exhibits an emerging covalent character, although the ionic character predominates. The charge excess is accumulated mostly in the bond regions, and not in the oxygen nuclei, a phenomenon observed in both adsorbates.

### Limitations and perspectives

This study involves some restrictions, since the calculations were carried out for an ideal monolayer of IT-MnO$_2$ in a vacuum and in base state (T→0 K, P ≈ ±1 kbar), a setting that differs a little from the experimental conditions where this type of material could exhibit some surface defects or a solvent environment. Although the idealization of the system allows an exhaustive, controlled analysis, we recognize that vacancies and impurities, solvents and humidity can modify mainly the structural and/or electronic properties of the IT-MnO$_2$ system, which could affect the results of the adsorption of atomic or molecular adsorbates. In spite of the restrictions involved in our calculations, the results obtained in this investigation afford us valuable and reliable information about the mechanisms of adsorption of atoms of toxic heavy metal atoms onto a T-MnO$_2$ monolayer. The good agreement of our results with the existing experimental results [55], in particular in relation to the type of adsorption that occurs in both Pb/MnO$_2$ systems, chemical adsorption, support the reliability of the calculations, their relevance, and the prospective applications in the removal of environmental contaminants, although in future studies we should explicitly explore defects, functionalization, and the effects of solvents in order to close the gap between ideal models and models under more realistic conditions.

### 4. Conclusions

The structural and electronic properties, stability, and adsorption of Cd and Pb onto the 1T-MnO$_2$ monolayer were studied using first-principles calculations. We found that the PMn site is the most energetically favorable for the adsorption of a Cd or Pb atom, with adsorption energies of –0.883 eV and –5.918 eV, respectively. The Cd atom transfers a charge of

approximately 1 e⁻ to the monolayer, while Pb transfers around 1.4 e⁻; this charge is then redistributed among first- and second-neighbour oxygen atoms. Additionally, once adsorption is complete, the Cd and Pb atoms are bound to the monolayer through a hybrid ionic-covalent interaction.

In relation to the pristine monolayer, the calculated exfoliation energy was 16.925 meV/A, from which it can be deduced that the monolayer is easily exfoliable. The calculations of the energy formation (Ef = −5.741 eV), the elastic constant, and the vibrational properties show that the monolayer is thermodynamically, mechanically, and vibrationally stable. The monolayer exhibits semiconductor behavior, with indirect band gaps of 2.08 eV and 2.02 eV for the spin-up and spin-down components, respectively. Furthermore, it exhibits magnetic behavior, with a magnetic moment of $3\mu_\beta$/cell.

## Author contributions

**Conceptualization:** César Ortega-Lopez, Gladys R. Casiano-Jiménez, Miguel J. Espitia-Rico.

**Data curation:** Mario L. Arteaga-Calderón, Miguel J. Espitia-Rico.

**Formal analysis:** César Ortega-Lopez, Miguel J. Espitia-Rico.

**Investigation:** César Ortega-Lopez, Miguel J. Espitia-Rico.

**Methodology:** César Ortega-Lopez, Julieth V. Dita-Casiano, Gladys R. Casiano-Jiménez, Miguel J. Espitia-Rico.

**Software:** César Ortega-Lopez, Julieth V. Dita-Casiano, Mario L. Arteaga-Calderón, Miguel J. Espitia-Rico.

**Supervision:** César Ortega-Lopez, Julieth V. Dita-Casiano, Miguel J. Espitia-Rico.

**Writing – original draft:** César Ortega-Lopez, Miguel J. Espitia-Rico.

**Writing – review & editing:** Miguel J. Espitia-Rico.

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
