## [Decision Letter · Decision Letter 0]

4 Jun 2025

Dear Dr. ORTEGA LÓPEZ,

Thank you for submitting your manuscript to PLOS ONE. After careful consideration, we feel that it has merit but does not fully meet PLOS ONE’s publication criteria as it currently stands. Therefore, we invite you to submit a revised version of the manuscript that addresses the points raised during the review process.

We look forward to receiving your revised manuscript.

Kind regards,

Veer Singh, Ph.D

Academic Editor

PLOS ONE

Reviewers' comments:

Reviewer's Responses to Questions

**Comments to the Author**

1. Is the manuscript technically sound, and do the data support the conclusions?

Reviewer #1: Yes

Reviewer #2: Partly

Reviewer #3: Yes

2. Has the statistical analysis been performed appropriately and rigorously?

Reviewer #1: Yes

Reviewer #2: N/A

Reviewer #3: Yes

3. Have the authors made all data underlying the findings in their manuscript fully available?

Reviewer #1: Yes

Reviewer #2: Yes

Reviewer #3: Yes

4. Is the manuscript presented in an intelligible fashion and written in standard English?

Reviewer #1: Yes

Reviewer #2: No

Reviewer #3: Yes

Reviewer #1: Authors have written the manuscript in a well-mannered way with all possible explanations. The introduction has raised several questions regarding the lack of studies related to the computational studies. Authors are asked to include those related studies in the introduction section.

Reviewer #2: The manuscript titled ‘First-principles calculation of adsorption Cd and Pb on the surface of a 1T-MnO₂ monolayer’ is well-structured and presents an interesting approach. However, several aspects require significant improvement. I recommend revisions to enhance its clarity, consistency, and overall impact.

• The abstract outlines a relevant and technically valuable study; however, it requires major revision to improve clarity, fix grammatical errors, and correct awkward phrasing such as “adsorption the toxic heavy atoms,” as well as repeated wording like “of the of the.”

• Terms like PMn and others must be clearly defined at their first use throughout the manuscript.

• Correct typographical errors, such as changing “Quatum-Espresso” to “Quantum ESPRESSO,” to ensure accuracy and consistency throughout the manuscript.

• “T-MnO₂” and “1T-MnO₂” are used interchangeably without clarification.

• The conclusion section should be revised to be more concise.

Reviewer #3: Regarding Computational Methodology:

Could the authors provide more details about the specific exchange-correlation functional used within DFT? Different functionals can influence the calculated adsorption energies and charge transfer. A justification for the chosen functional's suitability for this system would be beneficial.

What was the size of the 1T-MnO2 monolayer used in the calculations, and were any convergence tests performed to ensure that the size of the monolayer was sufficient to accurately model the adsorption of Cd and Pb? Providing information about the supercell size and k-point sampling is crucial.

How were van der Waals (vdW) interactions accounted for in the calculations? vdW forces can play a significant role in the adsorption of heavy metals on surfaces, and their inclusion (or lack thereof) should be clearly stated and justified.

Were spin-polarized calculations performed? Given that MnO2 can exhibit magnetic properties, it is important to clarify whether spin polarization was considered and how it affected the results.

Regarding Results and Discussion:

The adsorption energies indicate a strong binding of Pb compared to Cd. Could the authors provide a more in-depth analysis of the electronic structure changes upon adsorption to explain this difference? For example, a comparison of the density of states (DOS) or partial density of states (PDOS) for Cd and Pb adsorption would be helpful.

The Bader charge analysis suggests ionic bonding. Could the authors supplement this analysis with a discussion of the bond lengths and bond angles between the adsorbed atoms (Cd and Pb) and the substrate atoms (Mn and O)? This would provide further evidence to support the nature of the bonding.

The manuscript mentions that the 1T-MnO2 monolayer is "easily exfoliable." How does the calculated exfoliation energy compare to other known 2D materials? Providing a comparative context would strengthen this claim.

While the pristine 1T-MnO2 monolayer's stability is discussed, how does the adsorption of Cd and Pb affect the stability of the overall system? Were any calculations performed to assess the stability of the Cd/MnO2 and Pb/MnO2 systems (e.g., phonon dispersion calculations for the adsorbed systems)?

The conclusion mentions that the pristine monolayer exhibits semiconductor behavior. How does the adsorption of Cd and Pb affect the electronic properties of the 1T-MnO2 monolayer (e.g., changes in the band gap, metallization)? This is crucial for understanding the potential applications of this material in heavy metal removal.

Regarding Broader Context and Implications:

How do these calculated adsorption energies compare to experimental values (if available) or to other theoretical studies on similar materials? A comparison with existing literature would help to validate the accuracy of the calculations.

What are the potential limitations of this study? For example, the calculations are performed for an ideal 1T-MnO2 monolayer in vacuum. How might the presence of defects, surface functionalization, or a solvent environment affect the adsorption behavior in real-world applications?

What are the implications of these findings for the design of materials for heavy metal removal? Could the authors discuss the potential advantages and disadvantages of using 1T-MnO2 monolayers compared to other materials?

I will recommend authors to use (https://doi.org/10.1093/narmme/ugae005 and https://doi.org/10.1021/acsmedchemlett.3c00537 and https://doi.org/10.3390/scipharm93010006 references in introduction.

**Do you want your identity to be public for this peer review?** For information about this choice, including consent withdrawal, please see our Privacy Policy

Reviewer #1: **Yes: ** RAHUL RANJAN

Reviewer #2: No

Reviewer #3: **Yes: ** THUMPATI PRASANTH

---

## [Author Response · Author response to Decision Letter 1]

30 Jun 2025

Dear editor PLOS ONE Journal

We have made the suggested corrections, which are marked in red in the manuscript.

Editor and Reviewer comments:    

Reviewer 1

Comments

The manuscript titled ‘First-principles calculation of adsorption Cd and Pb on the surface of a 1T-MnO₂ monolayer’ is well-structured and presents an interesting approach. However, several aspects require significant improvement. I recommend revisions to enhance its clarity, consistency, and overall impact.

1. The abstract outlines a relevant and technically valuable study; however, it requires major revision to improve clarity, fix grammatical errors, and correct awkward phrasing such as “adsorption the toxic heavy atoms,” as well as repeated wording like “of the of the.”

R/ We have made the corrections.

2. Terms like PMn and others must be clearly defined at their first use throughout the manuscript.

R/ The terms PMn and others are defined in section 3.2.1. in red text.

To determine the most favorable adsorption models of Cd and Pb onto the 1T-MnO2-3x3 monolayer, we considered five (5) non-equivalent special sites, above an oxygen atom O ( site PO1): in the upper plane oxigen atoms; above a manganese atom Mn (site PMn): in the intermediate plane of manganese atoms; above an oxygen atom O (site PO2): in the lower plane, and above the Br1 bridge (site PB1), at the midpoint between an oxygen atom of the upper plane and an oxygen atom of the lower plane, and above Br2 bridge (site PB2): at the midpoint between an oxygen atom of the upper plane and a manganesse atom of the intermediate plane, as shown in Figure 6.

3. Correct typographical errors, such as changing “Quatum-Espresso” to “Quantum ESPRESSO,” to ensure accuracy and consistency throughout the manuscript.

R/ We have made the corrections

4. “T-MnO₂” and “1T-MnO₂” are used interchangeably without clarification.

R/ We would like to express our embarrassment for having made a notation error, since the correct notation is 1T-MnO₂. We have made the corrections.

5. The conclusion section should be revised to be more concise.

R/ The conclusion is now:

In summary, the structural and electronic properties, the stability, and the Cd and Pb adsorption on the 1T-MnO2 monolayer were studied using first-principles calculations. We found that the PMn site is the most energetically favorable for the adsorption of a Cd or Pb atom, with adsorption energies of – 0.883 eV and -5.918 eV, respectively. The Cd atom transfers a charge of 1e- to the monolayer, and the Pb transfers 1.4e. Aditionally, once the adsorption is complete, the Cd and the Pb atoms are bound to the monolayer by means of an ionic bond.

In relation to the pristine monolayer, the calculated of the exfoliation energy is 16,925 meV/A, from which it can be deduced that the monolayer is easily exfoliable. The calculations the energy formation (Ef = -5.741 eV), the elastic constant and vibrational properties show that the monolayer is thermodynamic, mechanic and vibrationally stable. The monolayer exhibits semiconductor behavior, with indirect band gaps of 2.08 eV and 2.02 eV for the spin-up and spin-down components, respectively. Furthermore, it exhibits magnetic behavior, with a magnetic moment of 3μβ/cell.

Reviewer 2

Comments

Abstract:

1. the charge transfer in the adsorbate/monolayer interaction 1T-MnO2 is established via the Bader charge. Please correct “the” to “The”.

R/ We have made the corrections

2. Please avoid using abbreviations in the keyword and make it short too.

R/ We have made the corrections

3. The MnO2 monolayer crystallizes in the trigonal (denoted 1H) phase with the D3h point group and the octahedral (denoted 1T) with D3d point group. Please explain D3h and D3d point groups.

R/ In the introduction section, we have added a brief description to explain D3h and D3d point groups, and we have added the text in red:

The MnO2 monolayer crystallizes in the trigonal (denoted 1H) phase with D3h point group (in the Hermann-Mauguin notation, D3h = 6 m2, where 6 is called the rotation-inversion axis (improper rotation) and consists of a rotation of order 6 (2π/6 rad=π/3 rad=60°) followed by a 1 inversion equivalent to a plane of symmetry or mirror, m is the vertical plane of symmetry (parallel to the main axis), and 2 the rotation axis of order 2 (2π/2 rad=π rad=180° perpendicular to the principal axis), and the octahedral (denoted 1T) with D3d point group (in the Hermann-Mauguin notation, D3d = 3 m, where 3 is called the rotation-inversion axis (improper rotation) and consists of a rotation of order 3 (2П/3rad=120°) followed by an inversion 1 (equivalent to a plane of symmetry or mirror); m is the symmetry plane (diagonal)).

4. The introduction needs to be improved. It is too short and includes examples of computational studies.

R/ We have added more computational studies, and five more references related to theoretical studies. In the paragraph we added, the text is in red.

Recent experimental studies reveal that the manganese dioxide monolayer is an excellent material for the removal of atoms, molecules, and gases that are very dangerous global pollutants due to their high toxicity. Wu et al. [21] demonstrated that sulfur dioxide (SO2) is efficiently captured via heterogeneous oxidation into sulfate on the surface of manganese dioxide (MnO2) with an efficiency in the capture of nearly 100%. Liu et al. [22] developed a filter to capture SO2 produced in the combustion of fossil fuels and released into the environment by vehicles and industrial smokestacks. Peng et al. [23] demonstrated the great potential of the MnO2 monolayer for the electrocatalytic transformation of CO2 to valuable chemicals products, because they proved that manganese dioxide reduces CO2 to CO with an efficiency of 71%. This is great importance for the chemical industry, because it constitutes an alternative path for the synthesis of important chemical feedstocks and complex carbon-based fuels. On the other hand, theoretical studies predict the use of the MnO2 monolayer in different areas. Deng et al. [24], using first-principles calculations, showed that it exhibits good performance for Li storage capacity and diffusion, and therefore the monolayer is promising electrode material for high-capacity Li ion batteries. Chen et al. [25], using density functional calculations, studied the adsorption of 5-fluorouracil (5-FU) on the surface of MnO2 and predicted that the monolayer is a good material for targeted delivery of drug molecules hiking on nanomaterials. Additionally, other theoretical studies based on the framework of density functional theory have focused on the adsorption of toxic atoms and molecules that cause environmental pollution. Independently, Zhu et al. [26] and Li et al. [27] studied the adsorption of NO and O2 molecules on the MnO2 surface, finding that this material is an excellent candidate for reducing emissions of these gases. Zhen at al. [28] calculated the adsorption of elemental mercury (Hg0) on MnO2; meanwhile, Zhang et al. [29] studied the adsorption mechanism of mercury species (HgO, HgCl, and HgCl2) on the MnO2 surface. These theoretical investigations predicted that MnO2 is a good material for the adsorption and oxidation of the Hg0 and mercury species HgO, HgCl, and HgCl2, and therefore MnO2 is a very promising material for the construction of devices (filters) that allow reducing emissions of mercury and mercury species to the environment that result from the burning of fossil fuels from power plants and vehicle exhausts [30].

5. Authors are advised to include the figure just above the figure caption for better understanding.

R/ We have included the figures in the manuscript.

6. Figure 4. Contour map of the charge distribution on the T-MnO2 monolayer, side view. The red color indicates little or no charge, and blue indicates high charge presence. The atoms of the upper and lower plane are of oxygen, and those of the central plane are of manganese. Please increase the font size of Figure 4.

R/ Thank you very much for the suggestion. We have increased the size of Figure 4.

Overall comment: The manuscript is well written and explained. However, some changes are required in the introduction section and the grammar of the manuscript needs to be checked once again. The manuscript can be published if

---

## [Decision Letter · Decision Letter 1]

1 Aug 2025

Dear Dr. ORTEGA LÓPEZ,

Thank you for submitting your manuscript to PLOS ONE. After careful consideration, we feel that it has merit but does not fully meet PLOS ONE’s publication criteria as it currently stands. Therefore, we invite you to submit a revised version of the manuscript that addresses the points raised during the review process.

We look forward to receiving your revised manuscript.

Kind regards,

Veer Singh, Ph.D

Academic Editor

PLOS ONE

Journal Requirements:

Additional Editor Comments:

Dear Author,

The manuscript titled "First-principles calculation of adsorption Cd and Pb on the surface of a 1T-MnO2 monolayer” is found it to be interested and informative work. However, I have few suggestions for minor revisions that I believe will further enhance the clarity and impact of your manuscript:

Please do not bold equations mentioned in the manuscript (e.g. 3.1, 3.2……..).

Please make sure to define each acronym at its first use. Check through the entire manuscript to make sure it is defined at the first use.

Grammar and Punctuation: There are several grammatical and punctuation errors throughout the manuscript.

Please add limitation of study and future direction as separate section in the manuscript. Please do not use unrelated citations suggested by reviewers.

Reviewers' comments:

Reviewer's Responses to Questions

**Comments to the Author**

Reviewer #1: All comments have been addressed

Reviewer #2: All comments have been addressed

Reviewer #3: All comments have been addressed

2. Is the manuscript technically sound, and do the data support the conclusions?

Reviewer #1: Yes

Reviewer #2: Yes

Reviewer #3: Yes

3. Has the statistical analysis been performed appropriately and rigorously?

Reviewer #1: Yes

Reviewer #2: Yes

Reviewer #3: Yes

4. Have the authors made all data underlying the findings in their manuscript fully available?

Reviewer #1: Yes

Reviewer #2: Yes

Reviewer #3: Yes

5. Is the manuscript presented in an intelligible fashion and written in standard English?

Reviewer #1: Yes

Reviewer #2: No

Reviewer #3: Yes

Reviewer #1: Authors have addressed all the comments in a well-mannered way and followed all the guidelines of the journal. Now it can be published in PLOS one.

Reviewer #2: The authors have thoroughly addressed all previously raised concerns, including clarifications in terminology, grammatical corrections, and improvements in technical writing.

Reviewer #3: The authors did not included the references I have selected, and therefore please review all of my comments address them individually

**Do you want your identity to be public for this peer review?** For information about this choice, including consent withdrawal, please see our Privacy Policy

Reviewer #1: **Yes: ** RAHUL RANJAN

Reviewer #2: No

Reviewer #3: **Yes: ** THUMPATI PRASANTH

---

## [Author Response · Author response to Decision Letter 2]

4 Sep 2025

Responses to the editor's comments and reviewers questions were all answered in the file: Response to Reviewers

---

## [Editor Report · Decision Letter 2]

7 Oct 2025

First-principles calculation of adsorption of cadmium and lead on the surface of a 1T-MnO2 monolayer

PONE-D-25-19623R2

Dear Author,

We’re pleased to inform you that your manuscript has been judged scientifically suitable for publication and will be formally accepted for publication once it meets all outstanding technical requirements.

Kind regards,

Veer Singh, Ph.D

Academic Editor

PLOS ONE

Additional Editor Comments (optional):

Dear Author,

The author has carefully revised the manuscript and provided detailed responses to all reviewer comments. The revised version adequately addresses the concerns raised during the review process. Therefore, I recommend the manuscript for publication.
---

## [Editor Report · Acceptance letter]

PONE-D-25-19623R2

PLOS ONE

Dear Dr. Ortega-Lopez,

I'm pleased to inform you that your manuscript has been deemed suitable for publication in PLOS ONE. Congratulations! Your manuscript is now being handed over to our production team.

Kind regards,

on behalf of

Dr. Veer Singh

Academic Editor

PLOS ONE